# Simulation and Key Physical Drivers of Primary Productivity in a Temperate Lake during the Ice-Covered Period: Based on the VGPM Model

Jie Zhang [1], Fei Xie [2], Haoming Song [1], Jingya Meng [1] and Yiwen Zhang [1,*]

[1] School of Ocean Science and Technology, Dalian University of Technology, Panjin 124221, China
[2] State Key Laboratory of Coastal and Offshore Engineering, Dalian University of Technology, Dalian 116024, China
* Correspondence: zhangyiwen@dlut.edu.cn

**Abstract:** The primary productivity of seasonal ice-covered water bodies is an important variable for understanding how temperate lake ecosystems are changing due to global warming. But there have been few studies on the complete change process of primary productivity during the ice-covered period, and the connection between ice physical and associated biological production has not been fully understood. In this study, a Vertically Generalized Production Model (VGPM) suitable for the ice-covered period was used to calculate the primary productivity of a temperate lake, and the key physical controlling factor was analyzed in the process of primary productivity change in the ice-covered period. The results showed that there was a high level of primary productivity, $(189.1 \pm 112.6)$ mg C·m$^{-2}$·d$^{-1}$, under the ice in the study site, Hanzhang Lake. The phytoplankton production under the ice was not as severely restricted by light as commonly thought. The water temperature played a more crucial role in the changes of primary productivity than the light beneath the ice. The study highlighted the variability in primary productivity covering the whole ice-covered age, and provided a better understanding of how the aquatic environment of lakes in seasonal ice-covered areas was affected by warmer temperatures.

**Keywords:** ice-covered period; primary productivity; temperate lakes; VGPM model; climate change

## 1. Introduction

Temperate lakes with seasonal freezing are considered some of the most sensitive regions to global warming [1,2]. The warming of lakes affects the growth and melting of ice sheets [3]. As the ice cover weakens, the light increases and water temperatures rise, which lead to an increase in the activity of organisms living under the ice [4]. In recent years, researchers have found that the subglacial water bodies can be quite productive in winter months, despite the cold temperatures and lack of light, which breaks with the past [5–8]. Another reason why biological activity under the ice is crucial, is that the lake ecology will change with the alternation of the four seasons in temperate lakes. Each season has its own characteristics that can affect the lake's ecosystem, and there is a close relationship between seasons. For example, the changes in biological activity with the melting of ice in winter can affect the distribution, structure, and biomass of organisms in the following spring [9–11]. Therefore, the primary productivity beneath the ice of seasonal ice-covered water bodies is an important factor for a comprehensive understanding of how temperate lake ecosystems are changing under the background of global warming.

The model estimation method is one of the main ways of calculating primary productivity. This method involves using a mathematical or simulation model to estimate productivity based on various parameters, including temperature, light, and chlorophyll over a wide range of timescales. The concentration of chlorophyll on the water surface can be determined by remote sensing technology, and the Vertically Generalized Production

Model (VGPM) is a model used to estimate the primary productivity of the water bodies based on chlorophyll concentrations [12,13]. Lomas [14] confirmed that the VGPM model can accurately estimate primary productivity by comparing measurement of chlorophyll *a* (Chl.a) with radioactive $^{14}$C isotope. The depth of the euphotic layer (which is the part of the water column where light can penetrate) is an input of the VGPM model, as it affects the amount of light that can reach the bottom layers of the water column. Generally, the depth at which 1% of the Photosynthetically Active Radiation (PAR) is taken on the water surface as the standard is typically used to determine the euphotic depths, or the photosynthetically active layer [15]. Yu calculated the primary productivity [16] below the ice by the amount of PAR that reaches the ice-water interface. The studies mentioned above have provided a foundation for the research into primary productivity in lakes with ice cover. But there have been relatively few studies on the complete change process of primary productivity during the ice-covered period. It is important to understand the primary productivity throughout the ice-covered period in order to inform our understanding of how aquatic ecosystems are responding to environmental changes. In addition, the connection between ice physical processes and ice ecosystem has not been fully established. For example, water temperature affects the vertical exchange of water [17], and the shortening of ice growth and melting lead to increased light availability under the ice, which enhance biological activity [18]. However, the specific effects of these physical factors on primary productivity remains to be studied. In this study, a VGPM model suitable for the ice-covered period was used to calculate the primary productivity and analyze key physical controlling factors, in order to establish the relationship between ice physics and ice ecology during the ice-covered period of a seasonal frozen temperate lake.

## 2. Materials and Methods

### 2.1. Study Site

Hanzhang Lake (40°40′ N–40°43′ N, 122°0′ E–122°08′ E) is located at the northern boundary of offshore China (Figure 1a), adjacent to Bohai Sea, with a salinity of 5–7 ppt. The lake has a surface area of 10 km$^2$ and it is generally shallow with an average depth of 6 m and a maximum depth of 10 m. The climate in this location is warm temperate continental semi-humid monsoon, with sufficient sunshine, and an average annual temperature of 10.5 °C. Hanzhang Lake generally enters the freeze period in December and melts in March of the next year, with a maximum ice thickness of 40 cm. The average winter temperature is approximately −6 °C, and the lowest temperature can reach −22 °C. The lake is eutrophic with an average total nitrogen (TN) concentration of 1.15 mg/L and an average total phosphorus (TP) concentration of 0.16 mg/L between 2019 to 2021.

### 2.2. Field Methods

The field study was conducted from 12 January to 14 March 2022, when the lake was covered by ice. The study was divided into two parts: floating remote observation platform and manual investigation. Ice thickness, PAR, water temperature, turbidity, and Chl.a were automatically monitored by sensors mounted on a floating remote observation platform (Figure 1b) [19]. The sensors used were: ultrasonic rangefinders with an accuracy of ±0.01 m and a monitoring frequency of every 1 min, solar radiation sensors with a monitoring frequency of every 30 min (two sets of sensors on the ice surface, measuring the incident light irradiance and the reflected light irradiance of the ice surface, respectively; one was set at 0.8 m below the ice), and YLS-ZDW chlorophyll and turbidity in situ monitoring sensors (the water temperature measurement range was −5 °C to 50 °C, with a resolution of 0.01 °C and an accuracy of ±0.15 °C; the turbidity range was 0 to 1000 FTU, with a resolution of 0.01 FTU and an accuracy of ± 2%; the range of Chl.a was 0 to 400 μg/L, with a resolution of 0.01 μg/L and an accuracy of ±5%; the monitoring frequency was once every 1 min; there were four sets at the water depth of 0.7 m, 1.5 m, 2.0 m, and 4.4 m, respectively). The transparency of water was measured by manual investigation with a Sayer's plate. Due to safety problems, the manual sampling time was based on the ice

thickness, starting at 15 cm during the freezing period and ending at 15 cm during the melting period. The sampling time interval was 5 days.

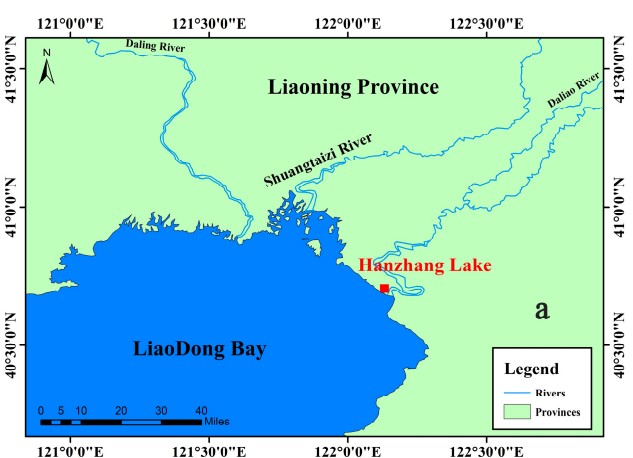

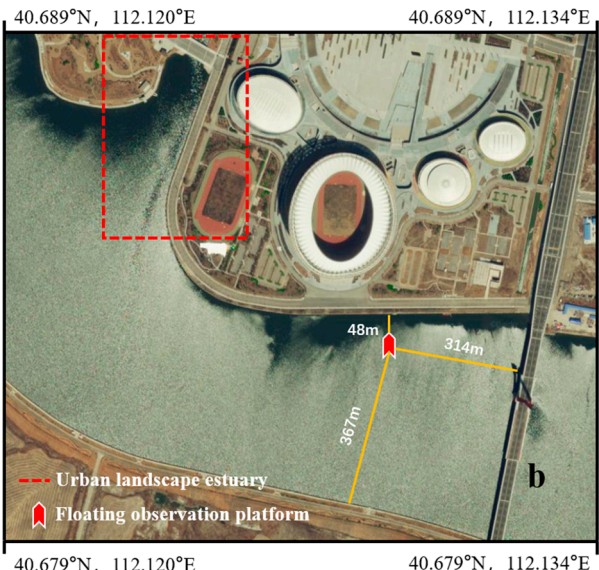

**Figure 1.** Study site and the floating observation platform ((**a**) Hanzhang Lake sampling point; (**b**), floating observation platform). The study sites are marked by solid lines.

*2.3. Euphotic Depths*

The calculation of euphotic depth ($Z_{eu}$) in this study refers to the following improved method [16]. Generally, non-icy water was considered to have uniform properties. According to Lambert-Beer law, the PAR of a beam of light passing through water will decrease exponentially with increasing depth. The euphotic depth was calculated by the Formula (1), the maximum depth at which light is still able to penetrate the water, allowing photosynthesis to occur:

$$Z_{eu} = \frac{2ln10}{K_d(PAR)} = \frac{4.605}{K_d(PAR)} \tag{1}$$

where $K_d$ *(PAR)* was the diffuse attenuation coefficient of photosynthetically active radiation ($m^{-1}$). Holmes [20] defined the relationship between transparency and $K_d$ *(PAR)* as:

$$K_d(PAR) = \frac{f}{SD} \tag{2}$$

where *SD* was the water transparency (in meters), and *f* was a constant that was determined by empirical data. Ma [21] investigated and analyzed data from 20 lakes in northern China

and found that the relationship between *Kd(PAR)* and *SD* was relatively stable, with a generally accepted value of 1.4 for the constant *f*. It was appropriate to select an *f* value of 1.4 for this study.

In frozen lakes during the winter months, ice and snow covered the water bodies, which can absorb and scatter solar radiation, thus affecting the light radiation through water bodies and altering the calculated euphotic depth. Therefore, the extinction coefficient of the ice sheet was a necessary factor for calculating the euphotic depth of water bodies in the ice period. PAR decreased exponentially in the ice sheet, and when combined with the mirror reflection of light on the ice surface, the extinction coefficient of the ice sheet was expressed as:

$$K_i(PAR) = \frac{1}{h_i}\left[(1 - R_s)\frac{E_d(0, PAR)}{E_d(h_i, PAR)}\right] \tag{3}$$

In general, the euphotic depth was defined as the depth where underwater photo-synthetically active radiation intensity was 14 $\mu$mol·m$^{-2}$·s$^{-1}$. The extinction coefficient and photosynthetically active diffuse attenuation coefficient of the ice sheet can be used to derive the calculation model of the euphotic depth of water in the ice-covered period:

$$Z_{eui} = \frac{SD}{f} ln \frac{(1 - R_s)E_d(0, PAR)}{14e^{k_i(PAR)h_i}} \tag{4}$$

where $Z_{eui}$ was the depth of the euphotic layer (m), *f* was 1.4, *SD* was water transparency (m), $h_i$ was ice thickness, $K_i$ *(PAR)* was the extinction coefficient of the ice layer (m$^{-1}$), $R_s$ was the reflectance of the surface mirror which was calculated by the ratio of surface reflection to surface incidence PAR, and $E_d$ (0, *PAR*) was the ice surface *PAR* ($\mu$mol·m$^{-2}$·s$^{-1}$).

### 2.4. Primary Productivity

The core calculation formula of primary productivity used in this study was the VGPM model established by Behrenfeld and Falkowski [22], based on large-scale and long-term monitoring data.

$$PP_{eui} = 0.66125 \times P_{opt}^B \times D_{irr} \times \frac{E_i}{E_i + 4.1} \times Z_{eui} \times C_{opt} \tag{5}$$

In the formula, $PP_{eui}$ was the primary productivity of the water body (measured by mgC·m$^{-2}$·d$^{-1}$) during the ice-covered period. $P_{opt}^B$ was the maximum carbon sequestration rate of the water column (mgC·mg$^{-1}$Chl·h$^{-1}$) during the ice-covered period. $D_{irr}$ is the illumination period (in hours) during the ice-covered period. $E_i$ was *PAR* at the interface of ice–water mixing (mol·m$^{-2}$·d$^{-1}$). In this study, $E_i$ was approximately replaced by *PAR* at 0.8 m water depth. $C_{opt}$ was the concentration of Chl.a ($\mu$g·L$^{-1}$) at the depth of the euphotic layer. In this study, an average value of Chl.a concentration at the water depths of 0.7 m and 1.5 m was used as an approximate replacement of that depth of euphotic layer.

The maximum carbon sequestration rate in the water column was calculated based on the Equation (6), proposed by Behrenfeld and Falkowski [22]:

$$\begin{aligned} P_{opt}^B = {} & 1.2956 + 0.2749T + 0.0617T^2 - 0.0205T^3 + 2.462 \times 10^{-3}T^4 \\ & - 1.348 \times 10^{-4}T^5 + 3.4132 \times 10^{-6}T^6 - 3.27 \times 10^{-8}T^7 \end{aligned} \tag{6}$$

where *T* was the water surface temperature (°C). There was a significant difference between the surface temperature in ice-covered and non-ice-covered water. In non-ice-covered water, the surface water temperature was usually the highest, and the maximum carbon sequestration rate of water column usually occurred at the surface water. However, for ice-covered waters, the water column temperature was lower, so the highest average temperature 2 m away from the water meter can be used as an approximate alternative.

The illumination period $D_{irr}$ indicated the daily length, which was the amount of time that the site was illuminated each day. The value can be obtained by querying geographic information about the study site.

### 2.5. Statistical Analysis

Stepwise linear regression and principal component analysis were used to identify statistically significant trends in the potential drivers of changing productivity. Statistical analysis was conducted using SPSS 9.0 and Origin 18.0. Data were considered significant when the difference was set at $p < 0.05$.

## 3. Results

### 3.1. Ice Thickness, Water Temperature, PAR, Chl.a, and Transparency

The ice thickness, water temperature, incident PAR, ice-water interface PAR, Chl.a, and transparency were shown in Figure 2, which were detected by sensors on the floating observation platform. The ice thickness increased during the freezing period, and reached a maximum peak of 0.43 m on 1 February. The ice water interface PAR fell to its lowest point due to snow, although the ice incident PAR was still increasing. This prevented light from entering the water, resulting in a decrease in the ice water interface PAR. As the temperature rose after 28 February, the ice began to melt gradually. The water temperature increased sharply from 2 °C to 4 °C during the end of February, and the lake ice entered a rapid melting period, melting at a rate of 2–3 cm per day until it had completely melted on 14 March. The incident irradiance showed an obvious increasing trend during the observation period as a result of an increase in both the total radiation and light time, because the time of the experiment was after the winter solstice and the direct solar point gradually moved northward from the Tropic of Cancer.

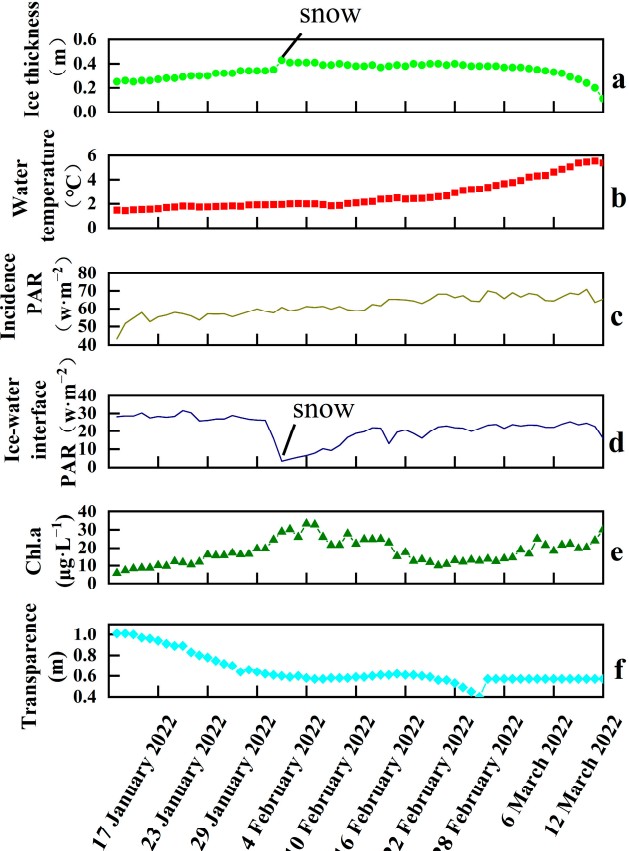

**Figure 2.** Changes of various environmental factors during the ice-covered period. ((**a**), Ice thickness; (**b**), water temperature; (**c**), ice incident PAR; (**d**), ice-water interface PAR; (**e**), Chl.a; (**f**), transparency).

The PAR at the ice-water interface was significantly lower than the incident irradiance on the ice surface and showed a trend of decreasing first before gradually increasing again. This was related to the formation and melting of ice. The ice thickness increased and enhanced the extinction effect during the freezing period. While the temperature rose and the solar radiation was further enhanced, the ice thinned, resulting in a compensation of the PAR at the ice-water interface in the melting period. The data showed that the average total PAR on ice surface was 61.73 W·m$^{-2}$, and the average PAR at the ice-water interface was 23.83 W·m$^{-2}$ without snow cover. The ratios of PAR at the ice-water interface and that at the ice surface were between 5.76% and 64.84%, respectively, with an average ratio of 35.69%.

During the whole period of ice cover, Chl.a showed a tendency to peak and then decrease. The average value of Chl.a was 17.89 μg·L$^{-1}$. The transparency was decreasing throughout the ice-covered period, and showed a negative correlation with Chl.a, indicating a relatively high level of phytoplankton productivity during the ice-covered period.

### 3.2. Primary Productivity

Primary productivity was calculated based on the data of ice thickness, water temperature, PAR, Chl.a, and transparency using the VGPM model, as shown in Figure 3. The primary productivity of Hanzhang Lake during the ice-covered period showed a trend of fluctuation increase. The minimum primary productivity was 57.77 mgC·m$^{-2}$·d$^{-1}$, which appeared on 12 January. The maximum value of primary productivity was 666.9 mgC·m$^{-2}$·d$^{-1}$, which appeared on 13 March. The mean value was (189.1.3 ± 112.6) mgC·m$^{-2}$·d$^{-1}$ during the whole ice-covered period. In the last days of the ice-covered period, the primary productivity increased rapidly and reached its maximum value.

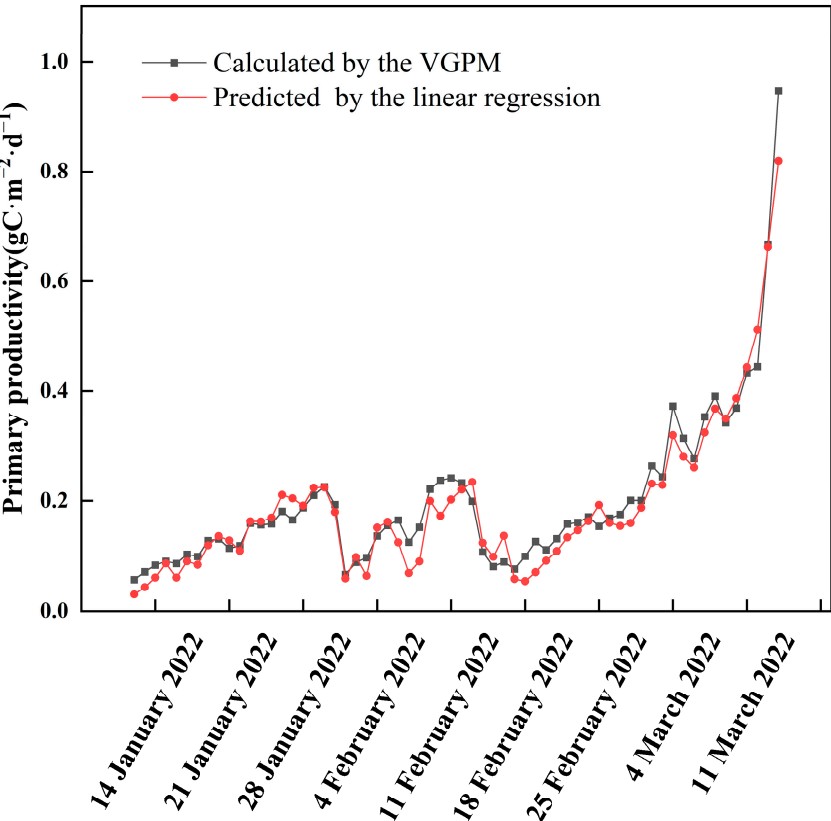

**Figure 3.** Primary productivity predicted by linear regression and calculated by the VGPM model.

A stepwise linear regression was used to figure out the importance of each calculation factor in the VGPM model, with transparency, ice thickness, ice surface incident PAR,

ice-water interface PAR, water temperature, and Chl.a as independent variables, and primary productivity as the dependent variable. The results showed that predicted values of linear regression were in good agreement with the calculated values of the VGPM model ($R^2 = 0.957$, $p < 0.001$). The linear regression formula was primary productivity = 0.242–0.775 × ice thickness (m)—0.358 × transparency (m) + 0.01 × chlorophyll a ($\mu g \cdot L^{-1}$) + 0.009 × daily average ice water interface PAR ($W \cdot m^{-2}$) + 0.021 × water temperature (°C), indicating that ice thickness, transparency, Chl.a, ice water interface PAR, and water temperature have a significant impact on primary productivity. The process of ice growth and melting, such as ice thickness, affected the dynamics of primary productivity under the ice.

### 3.3. Key Physical Factors

In terms of non-frozen open water bodies, there were great differences in physical factors such as light and water temperature of frozen water bodies, which can have an impact on the water ecosystem under ice. The water temperature usually affects phytoplankton photosynthetic enzyme activity and primary productivity. Changes in euphotic depth and PAR will be impacted by the light intensity, which in turn affects the photosynthetic intensity of phytoplankton, resulting in a change in primary productivity. Principal component analysis was performed on a variety of parameters including ice thickness, water temperature, ice incident PAR, ice-water interface PAR, Chl.a, and transparency in the VGPM model, as shown in Figure 4. Water temperature had the highest correlation with primary productivity throughout the ice period ($p < 0.05$ in the Bartlett test), suggesting that water temperature was a key factor in affecting primary productivity. At the same time, the entire ice period was divided into three parts, which was consistent with freezing and melting periods according to the ice thickness.

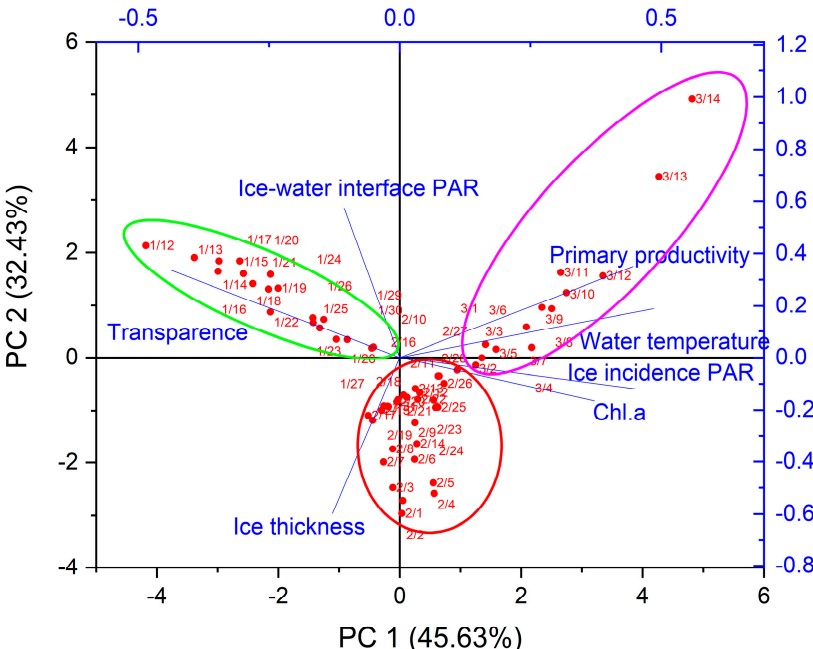

**Figure 4.** Principal component analysis of primary productivity and affecting factors in the VGPM model.

### 4. Discussion

The primary productivity of the Hanzhang Lake was 57.77–666.9 mg $C \cdot m^{-2} \cdot d^{-1}$ with an average of (189.1 ± 112.6) mg $C \cdot m^{-2} \cdot d^{-1}$. The result was comparable to that of other lakes or rivers calculated by the VGPM model (shown in Table 1), implying that there was a high level of primary productivity under the ice in Hanzhang Lake during the winter. The primary productivity of Hanzhang Lake was mainly attributed to

phytoplankton. Hanzhang Lake was meso-eutrophic with an average Chl.a concentration of $(18.4 \pm 7.21)$ µg·L$^{-1}$ and an average algae cell density of $(5.85 \pm 6.24) \times 10^6$ cells/L in water under the ice during the ice-covered period of 2022 [23]. The eutrophication of lakes changes the community structure of phytoplankton and benthic organisms significantly, which can lead to a shift in the water energy flow path from the bottom to the top [24,25]. As the phytoplankton was the main primary producer, the impact of environmental factors in the ice-covered period to the dynamics of phytoplankton was focused on in this study.

**Table 1.** Previous studies of primary productivity using the VGPM model.

| Lakes and Rivers | Latitudes | Time | Primary Productivity mg C·m$^{-2}$·d$^{-1}$ | References |
|---|---|---|---|---|
| Yenicaga Lake | 40°47′ N | December | 319 | [26] |
| Pearl River Estuary | 21°48′–22°27′ | Winter | 224.5 | [12] |
| Taihu | 30°55′40″–31°32′58″ N | Annual | 207.67–2237.71 | [27] |
| Tanganyika | 3°20′–8°48′ S | Wet and Dry Seasons | 110–1410 | [28] |
| Cape Fear River | – | Annual | 18–2580 | [29] |
| Wuliangsuhai | 40°36′–41°03′ N | Winter | 86.34–96.34 | [16] |

The effective light radiation under the ice was the primary limiting factor for aquatic biological activities [30]. Both the thickness and the structure of ice contribute to the light attenuation [31,32]. However, some studies suggest that the maximum photosynthetic efficiency required by phytoplankton was only 30% of that of benthic organisms [33]. Two pieces of evidence in this study suggested that the phytoplankton under the ice were not limited by light in Hanzhang Lake. First, it was found that primary productivity did not decrease with the decrease of light under the ice in the freeing period of this study, even though ice thickness and snow cover decreased the depth of the euphotic layer. Observations showed that the lowest depth of the eukaryotic layer was 0.8 m, providing plenty of space for phytoplankton to undergo photosynthesis. Second, the average daily ice-water interface PAR of the Hanzhang lake was $(21.7 \pm 6.9)$ W·m$^{-2}$, which met the lighting requirement for primary production. The quantum irradiance required for primary productivity was approximately 25 µmol·m$^{-2}$·s$^{-1}$, which was equivalent to approximately 5 W·m$^{-2}$ in terms of irradiance level [34]. Except for on overcast and snowfall days when light intensity decreased, primary productivity was observed to recover significantly with the melting of snow as PAR increased. This suggests that although PAR decreased due to snow and ice, it did not reach a level that limits primary production, possibly due to the geographic location of Hanzhang Lake.

This research suggests that water temperature is a key factor in determining the level of primary production in Hanzhang Lake, especially during periods of ice coverage. The data showed a statistically significant positive correlation between water temperature and concentration of chlorophyll a ($p < 0.05$), indicating that the warmer temperatures tend to lead to higher phytoplankton activities and primary production. The primary production of phytoplankton in the freezing period was mainly regulated by water temperature but not light. In addition, primary productivity increased quickly with the rise in temperature and improved light conditions during the period when the ice was melting. Water temperature and the duration of the ice cover were considered to be the main drivers of biological dynamics [35]. Aquatic conditions in winter played an essential role in the abundance and structure of phytoplankton communities, affecting the growth of phytoplankton during spring. Evidence showed change in phytoplankton in a long scale was linked to temperature rises, for example, phenology changes over a time period of 15 years in the Bassenthwaite Lake at the northern boundary of the English Lake District [1]. The dynamics and productivity of phytoplankton under the ice should be paid close attention to, in the background of global warming.

Convective mixing in the water layer beneath the ice played an important role in phytoplankton bioactivity, in addition to light and water temperature. This convective mixing

can create favorable conditions for bioactivity by increasing nutrient supply and allowing for more efficient exchanges between the surface and lower layers of water. Among the stratified water bodies of ice-covered closed freshwater lakes, the convective mixing in vertical profiles was usually weak. The salty lakes had a typical salt drainage effect when they froze, which can lead to a concentration and diffusion of the salty water layer beneath the ice. This can enhance the vertical convection of the water body. Hanzhang Lake was adjacent to the Bohai Sea, with a salinity of 5 to 7 ppt. The salt of the ice sheet would be transferred downward, and the nutrients released by the sediments can be more easily exchanged to the upper water when vertical convection was enhanced [36]. Thus, the nutrients in the water were accumulated, and the phytoplankton moved freely between the water layers, which contributed to the growth of phytoplankton populations. The light limitation below the euphotic layer was weakened when vertical convection was enhanced [34], which can be related to the high chlorophyll concentration in Hanzhang Lake during the ice-covered period. There was a negative correlation ($p < 0.05$) between Chlorophyll *a* and ice-water interface PAR in this study, and the high-value area of chlorophyll did not always exist in surface water. The negative correlation of Chl.a and PAR in Hanzhang Lake was consistent with that in Antarctic Lake Bonney [37], which was due to the self-shading effect. The shelf-shading effect was that high chlorophyll concentrations attenuated light radiation.

The results from Hanzhang Lake indicated that the primary productivity was greatly affected by the ice-covered period. Phytoplankton was the main producer, and water temperature was the key physical factor driving the dynamics of phytoplankton. The phytoplankton under the ice was not as light limited as commonly thought. The primary production level of phytoplankton will not decrease significantly, even though the thick ice and snow cover can lead to a decrease in light radiation and a decrease in the depth of the euphotic layer. This study was meant to provide reference for the subsequent evolution of ice ecological environment.

## 5. Conclusions

The ice-covered period of Hanzhang Lake lasted approximately 3 months, with a maximum ice thickness of 0.43 m and an average ice thickness of 0.29 m during winter in 2021–2022. The primary productivity was 57.77–666.9 mg C·m$^{-2}$·d$^{-1}$ with an average of (189.1 ± 112.6) mg C·m$^{-2}$·d$^{-1}$ during the ice-covered period, using the VGPM model. The water temperature played a more critical role in the changes of primary productivity than light under the ice in Hanzhang Lake. The primary production level did not decrease when the thick ice and snow cover led to the decrease of light radiation and the depth of the euphotic layer in the freezing period. In addition, primary productivity increased quickly with the rise in temperature and improved light conditions in the melting period. It appears that in regions where there are seasonal changes of icy cover, warmer water temperatures may be more important for primary productivity than light conditions, providing a better understanding of the interaction between ice physicals and ecology.

**Author Contributions:** Investigation, analysis, and writing of the original draft, J.Z.; investigation and conceptualization, F.X.; analysis and visualization, H.S.; formal analysis, J.M.; writing review and editing, Y.Z. All authors have read and agreed to the published version of the manuscript.

**Funding:** This research was supported by the National Key Research and Development Program of China (Grant No. 2019YFE0197600) and the National Natural Science Foundation of China (Grant No. 42007150, 52211530038).

**Institutional Review Board Statement:** No human or animal studies involved in this study.

**Data Availability Statement:** Data from the VGPM model in this study can be downloaded from https://pan.baidu.com/s/1Lg0Hqa2aoyFI_OUyr3-BYw (code 1952); the meteorological data from the National Meteorological Data Center can be downloaded from http://data.cma.cn.

**Acknowledgments:** We are grateful to the editor and anonymous reviewers for their comments, which considerably improved this work.

**Conflicts of Interest:** The authors declare no conflict of interest.

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
