# Peer review of "Simulation and Key Physical Drivers of Primary Productivity in a Temperate Lake during the Ice-Covered Period: Based on the VGPM Model"

_water, doi:10.3390/w15050918_

Round 1
Reviewer 1 Report
A field investigation was carried out in a temperate lake, and the primary productivity under the ice was calculated by the Vertically Generalized Production Model in the manuscript. The complete change process of primary productivity was analyzed over time during the ice-covered period. The findings showed that primary productivity was as high under the ice as that in open water conditions in the temperature lake, and the Vertically Generalized Production Model was able to accurately predict the levels of primary productivity. The experimental design is reasonable, and the data are reliable. English language and style are minor spell check required. The specific problems that need to be revised include correcting any spelling or grammar errors, ensuring the language and style of the paper is consistent and making sure the figures and tables are accurate and clear.
Line 53-54:Instead of starting a sentence with "Usually," it is best to use words like "generally" or "typically."
Line 83, Figure 1: The figure is not clear enough, especially Figure 1a. A graph with a resolution of 600 dpi should be provided to improve the clarity and accuracy of the study site. The higher resolution will allow more detailed information to be presented, which can help readers better understand the results of the experiment.
Line 110: Formula (1) - (6): Formulas should be centred on the line to make them easier to read and understand.
Line 158: The unit μg/L should be μg·L-1, consistent with others.
Line 200: Note decimal places and significant figures to ensure accuracy in the data that is being presented in the sentence.
Line 217: The stepwise linear regression with the SPSS 9.0 should be provided in the Materials and Methods section to help readers understand the statistical analysis used to come to the paper’s conclusions.
Line 237: The principal component analysis with Origin 18.0 should be provided in the Materials and Methods section.
Line 244: The symbols in Figure 4 overlap and adjusting is recommended.
Line 260-261: An updated reference requires citation about the effective light radiation under the ice and aquatic biological activities.
Author Response
Thank you for the reviewers comments concerning our manuscript. We have studied the comments carefully and revised manuscript according to the letter. Revised portion are showed in blue in the attachment.
Line 53-54:Instead of starting a sentence with "Usually," it is best to use words like "generally" or "typically."
Correct according to comments. "Generally" was used instead of "Usually" in the revision.
Line 83, Figure 1: The figure is not clear enough, especially Figure 1a. A graph with a resolution of 600 dpi should be provided to improve the clarity and accuracy of the study site. The higher resolution will allow more detailed information to be presented, which can help readers better understand the results of the experiment.
A revised figure clear enough was provided in the revision.
Line 110: Formula (1) - (6): Formulas should be centred on the line to make them easier to read and understand.
Correct according to comments. Formulas were centered on the line.
Line 158: The unit μg/L should be μg·L-1, consistent with others.
The unit μg/L was changed to μg·L-1.
Line 200: Note decimal places and significant figures to ensure accuracy in the data that is being presented in the sentence.
The decimal places and significant figures was revised to ensure the accuracy of data.
Line 217: The stepwise linear regression with the SPSS 9.0 should be provided in the Materials and Methods section to help readers understand the statistical analysis used to come to the paper’s conclusions.
Line 237: The principal component analysis with Origin 18.0 should be provided in the Materials and Methods section.
Part of Statistical analysis was added in the Materials and Methods section, including the methods and software.
Line 244: The symbols in Figure 4 overlap and adjusting is recommended.
The format of Figure 4 has been adjusted to meet the requirements.
Line 260-261: An updated reference requires citation about the effective light radiation under the ice and aquatic biological activities.
Updated references have been cited. A more balance and a better account of our work have been produced. I hope that the revised manuscript is now suitable for reading.

Reviewer 2 Report
The article addresses a very interesting issue (long underestimated) in limnological research which is the ecology during the winter season. The article compiles a wide range of parameters relating to the physical and biological characteristics of the lake in question. I recommend the article for publication after taking into account the comments below:
Materials and methods
I do not find information on the period of the research conducted
Results
Line 213-214: It is not possible to talk about the maximum productivity in the lake during the period of ice cover falling on March 14, since a few lines earlier the authors write that the ice had just disappeared on March 14 (Line 188).
The authors do not provide any information on the structure of the ice. Not only the thickness but also the composition (inclusions of solid facies, gas bubbles, etc.) of the ice sheet determine the possibility of light penetration. Please complete this information.
I would be cautious in making general conclusions based on a short campaign (2021-2022-this information is only found in the Conclusions section). In another season (i.e., with a different concentration of nutrients), will the situation be analogous to the current one?
Author Response
Thank you for the reviewers comments concerning our manuscript . We have studied the comments carefully and revised manuscript according to the letter. Revised portion are showed in blue in the attachment.
